# Time Series Ovarian Transcriptome Analyses of the Porcine Estrous Cycle Reveals Gene Expression Changes during Steroid Metabolism and Corpus Luteum Development

**DOI:** 10.3390/ani12030376

**Published:** 2022-02-04

**Authors:** Yejee Park, Yoon-Been Park, Seok-Won Lim, Byeonghwi Lim, Jun-Mo Kim

**Affiliations:** Functional Genomics & Bioinformatics Laboratory, Department of Animal Science and Technology, Chung-Ang University, Anseong 17546, Gyeonggi-do, Korea; tkfkdplus@cau.ac.kr (Y.P.); clickbyb@cau.ac.kr (Y.-B.P.); sw8333@cau.ac.kr (S.-W.L.); hwi1208@cau.ac.kr (B.L.)

**Keywords:** corpus luteum, differentially expressed gene analysis, estrous cycle, gene expression, ovarian transcriptome, pigs

## Abstract

**Simple Summary:**

The estrous cycle, which is divided into follicular and luteal phases based on ovulation, is influenced by reproductive hormones which affect reproduction and cause changes in the reproductive system of the pig. As the main reproductive organ, the ovary is involved in ovulation and changes in the corpus luteum. We aimed to identify dynamic changes in gene expression through differentially expressed gene profiling and to provide a comprehensive understanding of the molecular mechanisms that occur in the pig ovary during the estrous cycle. The transcriptome analysis revealed that the dynamic change in gene expression was more activated in the luteal phase than in the follicular phase. Functional analysis revealed that the metestrus and diestrus periods are important in preparation for pregnancy or the next estrous cycle after ovulation.

**Abstract:**

The porcine estrous cycle is influenced by reproductive hormones, which affect porcine reproduction and result in physiological changes in the reproductive organs. The ovary is involved in ovulation, luteinization, corpus luteum development, and luteolysis. Here, we aimed to provide a comprehensive understanding of the gene expression patterns in porcine ovarian transcriptomes during the estrous cycle through differentially expressed genes profiling and description of molecular mechanisms. The transcriptomes of porcine ovary were obtained during the estrous cycle at three-day intervals from day 0 to day 18 using RNA-seq. At seven time points of the estrous cycle, 4414 DEG were identified; these were classified into three clusters according to their expression patterns. During the late metestrus and diestrus periods, the expression in cluster 1 increased rapidly, and steroid biosynthesis was significant in the pathway. Cluster 2 gene expression patterns represented the cytokine–cytokine receptor interaction in significant pathways. In cluster 3, the hedgehog signaling pathway was selected as the significant pathway. Our study exhibited dynamic gene expression changes with these three different patterns of cluster 1, 2, and 3. The results helped identify the functions and related significant genes especially during the late metestrus and diestrus periods in the estrous cycle.

## 1. Introduction

The estrous cycle is a major regulatory factor of female fertility and influences the development of the reproductive organs, ovulation, and hormone secretion [1]. It affects litter size and the weight of offspring in the porcine industry [2,3]. Pigs are anatomically similar to humans, and genetic and physiological aspects are also similar to those of humans, so many studies and experiments are being conducted on pigs [4].The 21-day porcine estrous cycle is divided into the follicular and luteal phases based on the ovulation process, and involves various hormones such as gonadotropin releasing hormone, follicle stimulating hormone, luteinizing hormone (LH), progesterone, and estrogen [1]. Therefore, the female endocrine secretions in the estrous cycle are strongly related to the functional roles of the ovary.

The mammalian ovary is not merely a female reproductive organ where oocytes are stored and ovulation occurs, but it also regulates a variety of physiologically diverse aspects of reproduction [5]. After ovulation, the ovulated follicle turns into a corpus luteum (CL), which secretes reproduction-related hormones. Furthermore, the ovary produces steroid hormones (estrogen and progesterone) and peptide growth factors, which play key roles in the functioning of the ovary itself. These are also critical to the regulation of the hypothalamic-pituitary-ovarian axis and the development of secondary sex characteristics [1].

An ovarian transcriptome analysis has been conducted in recent years to identify differentially expressed genes (DEG). Some RNA-seq studies involving litter size in sows have been conducted to further porcine prolificacy [2,3]. A previous study of the ovary screened the major functional genes or molecular markers of estrus expression during diestrus and estrus in gilts from two different breeds [6]. In addition, previous research has provided comprehensive transcriptome data on the porcine ovary at the proestrus and estrus stages through RNA-seq technology [7]. Although we previously analyzed the whole estrous cycle using RNA-seq, we focused on the interaction and connectivity between three reproductive tissues (ovary, oviduct, and endometrium), rather than on examining the changes that occurred during the estrous cycle in a particular tissue [8]. Furthermore, few studies have been carried out on the entire gene expression changes during the development of follicles and CL in the porcine estrous cycle [8,9]. In this study, we aimed to provide a comprehensive understanding of the gene expression patterns in porcine ovarian transcriptomes during the entire estrous cycle through DEG profiling, cataloging, and description of molecular mechanisms.

## 2. Materials and Methods

### 2.1. Ethics Statement

All animal experiments were approved by the Institutional Animal Care and Use Committee of the National Institute of Animal Science, Republic of Korea (No. 2015-137). All experimental procedures were carried out in accordance with the Guide for Care and Use of Animals in Research and reported according to ARRIVE guidelines (https://arriveguidelines.org, accessed on 20 April 2017).

### 2.2. Animals and Sampling

We used 22 crossbred (Landrace × Yorkshire) gilts of approximately similar age (6–8 months) and weight (100–120 kg) who had undergone at least two normal duration estrous cycles. The pigs were selected from different pens from a single research farm in the National Institute of Animal Science (NIAS) to follow estrus behavior. Gilts were subject to estrus detection daily in the presence of boars; the first day of detected estrus behavior was designated as day 0. At days 0 (D00; n = 3), 3 (D03; n = 2), 6 (D06; n = 3), 9 (D09; n = 3), 12 (D12; n = 4), 15 (D15; n = 4), and 18 (D18; n = 3) gilts were sacrificed, and whole ovaries were dissected aseptically from the reproductive organs as previously described [10,11] in the euthanized gilts (Figure 1). The collected ovaries were ground whole and used as samples for the study. Ovarian tissues were snap-frozen in liquid nitrogen and stored at −80 °C for RNA extraction. These sampling materials were used in the previous study to integrate different kinds of female reproductive tissues and identify the core regulation module in the integrative gene expression networks [8].

### 2.3. RNA Extraction, Library Preparation, and Sequencing

Total RNA was extracted from the ovarian tissues using the TRIzol reagent (Invitrogen, Life Technology, Carlsbad, CA, USA) in accordance with the manufacturer’s recommendations. RNA integrity was assessed using electrophoresis in 1% agarose gel, and RNA with a RIN value of 7 or higher was used. The quantity was validated using the NanoDrop ND-1000 spectrophotometer (NanoDrop Technologies, Wilmington, DE, USA). Individual libraries were generated using the reagents provided in the Illumina TruSeq RNA sample preparation kit as described previously [12]. The sequencing was performed using the constructed cDNA library with the 100-base pair (bp) paired-end method on an Illumina HiSeq 2000.

### 2.4. Data Processing and DEG Identification

FastQC (v0.11.4) software was used for the quality checking tool of raw reads [13], and the reads were trimmed using Trimmomatic (v0.38) for adapters and low quality [14]. Hisat2 (v2.1.0) was utilized to map the clean reads against the porcine reference genome (*Sus scrofa* Sscrofa11.1.98, GCA_000003025.6) of the Ensembl genome browser (http://www.ensembl.org/Sus_scrofa/, accessed on 31 July 2020) as the default option of the program [15]. Samtools (v1.9) was used to sort the mapped reads and change the SAM file to a BAM file. The raw counts corresponding to the genes for each library were calculated based on exons in *Sus scrofa* GTF v98 (Ensembl) as the genomic annotation reference file using featureCounts (Subread package v1.6.3) [16]. The DEG analyses were performed at each time point (D03, D06, D09, D12, D15, and D18) and compared with D00, which resulted in six comparisons (D03 vs. D00, D06 vs. D00, D09 vs. D00, D12 vs. D00, D15 vs. D00, and D18 vs. D00). All DEG analyses for obtained raw counts were performed using the Bioconductor edgeR package v3.28.1 [17]. The genes were removed when all samples had raw counts of ≤10 to reduce statistical bias in the DEG analyses. Normalization for raw counts was performed using the trimmed mean of *M*-values method [18]. Limma, an R package, was used for performing MDS to identify the similarity among samples [19], and the ggplot2 R package was used for visualization of the MDS and volcano plot [20]. The DEG were identified based on a negative binomial generalized linear model, the *p*-value was adjusted using the Benjamini–Hochberg correction with a false discovery rate of <0.05, and absolute log_2_ fold change (FC) ≥ 1 was applied as the cutoff for DEG [21].

### 2.5. Functional Validation of DEG

Gene clustering analysis and visualization were conducted using Multi Experiment Viewer (MeV v4.9.0) to identify similarly expressed patterns of time-serial DEG [22]. Stringent thresholds (absolute log_2_ FC ≥ 2) for DEG were selected for the k-medians clustering analysis with 1K iterations. To annotate gene ontology (GO) terms and Kyoto encyclopedia of genes and genomes (KEGG) pathways using DEG corresponding to 7 time points and each cluster, the database for annotation, visualization, and integrated discovery (DAVID) v6.8 was used as a functional annotation tool [23]. Biological process (BP), cellular component (CC), and molecular function (MF) were included in GO terms. Next, REVIGO treemap [24] and ClueGo (v2.5.5) plugin of Cytoscape (v3.7.2) [25] were used to visualize BP, CC, and MF GO terms. A pie graph was composed with the pathways which met a threshold of *p*-value < 0.05, and the interval of GO tree was from the minimum level of 4 to the maximum level of 8. In addition, pathways with at least five genes were functionally grouped to define term–term interrelations based on their kappa score level (≥0.3). The treemap was modified to show representative terms among several similar terms, and the pie graph shows the proportion of each term in the functional analysis results. Subsequent analyses were performed using the DEG and functional annotation results on D06.

### 2.6. Gene Modulation in the KEGG Pathway Analysis

Considering the results of the BP terms in each cluster, functionally similar KEGG pathways were selected as a representative term. Next, the modulations of responsible genes (proteins) in the selected representative KEGG pathway through functional analyses were verified using the clusterProfiler R package [26]. Gene modulations were performed to identify how many core genes are expressed in the selected KEGG pathways and which parts of the pathway the gene is involved in. The genes with the greater log_2_ FC value are shown in dark red, whereas the genes with the lower log_2_ FC values are shown in dark blue in the selected pathway. Among the genes corresponding to each protein, genes indicating the absolute maximum change were used as a representative value. Additionally, the core enriched genes of the significant pathway were expressed as heatmaps.

## 3. Results and Discussion

### 3.1. Integrative RNA-seq Analysis and DEG Profiling during the Estrous Cycle

In the current study, we divided the porcine estrous cycle into seven time points based on the development of follicles (D00), ovulation (D03), luteinization and luteal development (D06–12), luteolysis (D15–18), and hormone secretions and generated RNA-seq data for each time point (Figure 1). For these seven time points, 430 million paired-end sequence reads were produced from 22 samples with an average of 19.5 million reads per sample. The average numbers of the clean reads after trimming were 16,428,136 (D00), 16,024,399 (D03), 17,400,840 (D06), 16,910,350 (D09), 15,876,680 (D12), 16,541,879 (D15), and 17,577,294 (D18). The average unique mapping rate (94.52%) and the average overall mapping rate (98.36%) were determined (Table 1). The RNA-seq data revealed a separation between the late metestrus and diestrus phases (D06, D09, and D12) and other phases (D00, D03, D15, and D18) in the multidimensional scaling (MDS) plot, which may indicate that important changes occurred during the late metestrus and diestrus phases compared to the other phases (Appendix A).

Identification of DEG in time series analysis is a key step towards better understanding the functional role of genes during the entire estrous cycle [27]. In our study, a total of 4414 DEG were identified from all time points based on D00 (Figure 2a and Appendix A). The number of DEG on D06 (n = 3538) and D09 (n = 3163) increased sharply in comparison to that on D03 (n = 965) and then decreased dramatically on D15 (n = 232) and D18 (n = 45) compared to that on D12 (n = 1683). This might be attributed to the fact that the oocyte-releasing follicles in the ovary develop into CL after ovulation (D06), which causes rapid physiological and morphological changes in the ovary [1]. Furthermore, a large proportion of downregulated DEG found on D06 corresponding to the metestrus, D09, and D12 corresponding to the diestrus phase were identified (D06: 55.90%; D09: 53.68%; D12: 52.88%). The overlapped DEG between D06 and D09 was the largest with 1167 genes (Appendix A).

A clustering analysis was applied to characterize the DEG according to similar expression patterns during the overall estrous cycle. As a result, the DEG were grouped into three clusters showing a similar pattern with dynamic expression changes (Appendix A). A total of 832 DEG were included in cluster 1, which was mainly upregulated on D06 and D09 (Figure 2b), whereas 1205 DEG in cluster 2 were involved with the downregulated pattern during the same period (Figure 2c). Cluster 3 included 436 DEG, which represented the only remarkable downregulation on D06 (Figure 2d). These clustered expression patterns indicated the processes of ovulation, luteinization, CL development, and luteolysis during the ovarian estrous cycle. In the process of CL development, apoptosis, angiogenesis, steroidogenesis, signal transduction, translation, cell proliferation, and tissue remodeling occur [28]. These findings show that marked changes in the ovarian transcriptome occur more in the luteal phase rather than in the follicular phase.

### 3.2. Functional Annotations and Enriched Pathway Analyses in Clusters

The GO and KEGG enrichment analyses were conducted to discover the connection between the estrous cycle and its molecular functions. The treemaps and pie graphs described the enriched terms in BP of GO, and the KEGG pathway terms were organized into bar plots (Figures 3, 5 and 7, and Appendix A). For BP terms, a total of three KEGG pathways were selected for each cluster (Supplementary Appendix A). Heatmaps were drawn to visualize specific DEG, based on D06, when the most dynamic change occurred in all three clusters (Figures 4, 6 and 8). This dynamic change in D06 could be attributed to the process of follicles transforming into CL after ovulation [1].

#### 3.2.1. Reproductive Hormones in Luteal Phase

In cluster 1, intestinal absorption and sterol biosynthesis were the most significant representative GO terms (Figure 3a), and similarly, sterol metabolic process and lipid biosynthetic process were the most significant terms in the pie graph (Figure 3b). As indicated in cluster 1, lipid metabolism (steroid biosynthesis), secretory and digestive system (bile secretion, pancreatic secretion, insulin secretion, gastric acid secretion, salivary secretion, and fat digestion and absorption), and amino acid (alanine, aspartate, and glutamate metabolism; valine, leucine, and isoleucine degradation; taurine and hypotaurine metabolism, and tyrosine metabolism)-related terms were the main pathways (Figure 3c). These significant terms were consistent with the changes that occur in hormones involved in the estrous cycle. Steroid biosynthesis related to steroid hormones and the digestive system including bile, pancreas, and insulin are also part of the lipid digestion process. In addition, they are directly and indirectly related to hormone secretion [29]. Several terms appear to display similar functions in the GO and KEGG results (sterol metabolic process, lipid biosynthetic process, and steroid biosynthesis). Based on BP terms, steroid biosynthesis was chosen as the major pathway in cluster 1. There were several KEGG orthologues related to steroid biosynthesis. Most of the upregulated genes shown in Figure 4a were DEG and were related to the enzymes and reactions for synthesizing cholesterol. Steroid biosynthesis in the ovary is very important for proper ovarian function. Previous studies have shown that rat androstenedione synthesis maintains the function of the CL and increases progesterone biosynthesis [30]. It has also been shown that androgens can maintain luteal function in mice lacking the classic progesterone receptor [31]. Therefore, it can be concluded that various steroid hormones are involved in steroid biosynthesis and progesterone formation in the CL. Progesterone especially is related to the morphological change from corpora lutea to corpus luteum [32]. In cluster 1, the fold change in the expression level of 10 steroid biosynthesis DEG (*HSD17B7*, *DHCR7*, *LSS*, *SQLE*, *SC5D*, *FDFT1*, *NSDHL*, *MSMO1*, *DHCR24*, and *TM7SF2*) were visualized at the time point of D06 through a heatmap (Figure 4b). *HSD17B7* participates in post-squalene cholesterol biosynthesis, which is a leading substance in steroids [33]. *LSS*, *SC5D*, *NSDHL*, *DHCR24*, and *TM7SF2* are involved in all stages of cholesterol biosynthesis, along with *SREBF1*, which is involved in lipid homeostasis and can be presumed to take part in steroid biosynthesis for pregnancy preparation during the late metestrus and diestrus (Figure 4a,b) [34].

Steroid hormones in the ovary are closely related to blood flow in the uterus; estrogen decreases blood flow, whereas progesterone inhibits its action [35]. Decreased blood flow in the uterus and the effect of hormones decreases the quality of oocytes, which causes infertility [36]. Therefore, proper blood supply to the uterus indicates endometrial acceptability and is associated with successful fertilization [37]. Late metestrus to diestrus (D06–D09), corresponding to all clusters, are the periods when CL development is most active, resulting in increased concentrations of progesterone. From this, it can be hypothesized that the expression of genes selected for DEG during this period had consequences associated with the development of the CL at the time and consequently an increase in progesterone concentrations. Progesterone also suppresses the development of new follicles by inhibiting the activity of the hypothalamus and the pituitary gland. Owing to these processes, the recurrence of the estrous cycle is controlled during the luteal phase or gestational phase. Next, in the latter part of diestrus (approximately D12), luteolysis begins if fertilization has not occurred. In addition, this might reflect that the amplitude of LH secretion in the luteal phase is similar to that in the gestational phase [1]. This implies that during the diestrus phase before luteolysis, the ovary forms substantial structures similar to those in the gestational phase for the preparation of pregnancy. As part of the preparation, the CL synthesizes progesterone, which is essential for the establishment and maintenance of early pregnancy. Therefore, the CL secretes progesterone to maintain its structure and continues to secrete progesterone during the maintenance process to repeat and maintain the process. In addition, progesterone prevents the degeneration of CL and maintains its structure during pregnancy [31].

#### 3.2.2. Immune Responses in Luteal Phase

The greatest enriched GO term for cluster 2 was neutrophil chemotaxis (Figure 5a). Immune-related terms, such as regulation of immune response and phagocytosis, constitute a large proportion of enriched GO terms (Figure 5b). KEGG pathways involved in immune disease (rheumatoid arthritis, asthma, autoimmune thyroid disease, allograft rejection, inflammatory bowel disease, graft versus host disease, and hematopoietic cell lineage), infectious disease (*Staphylococcus aureus* infection, leishmaniasis, and tuberculosis), and signaling molecules and their interactions (cytokine–cytokine receptor interaction, cell adhesion molecules, and neuroactive ligand–receptor interaction) were enriched in cluster 2 (Figure 5c). The KEGG pathway cytokine–cytokine receptor interaction was chosen as the significant pathway in consideration of BP terms in cluster 2. There are several KEGG orthologues related to cytokine–cytokine receptor interaction. One-third of these are expressed, and most downregulated genes were differentially expressed as shown in Figure 6a. Ovulation occurs at approximately D03, and the ovulated oocyte undergoes a chemical reaction with the sperm within two or three days, resulting in the establishment of pregnancy. It appears that immune-related pathways are downregulated to accept cells from other individuals to prepare for pregnancy until luteolysis begins around D12. Sperm influx causes the female immune system to recognize them as a foreign antigen and respond; however, the immune response at that time is downregulated [38,39]. It is postulated that the organs of the same reproductive system are organically connected, so that the immune system of the ovary may also change under the influence of the oviduct that the sperm cells enter. Cytokines stimulate and inhibit the development of steroids by ovarian cells, and particularly, inflammatory cytokines can regulate the number of leukocytes and the function of the endocrine system in the ovary. In addition, cytokines act on follicle growth, activation of leukocytes, ovulation, luteinization, and luteolysis and are also involved in both inhibition and stimulation of follicular responses to gonadotropin [40]. As important mediators of the immune response, cytokines are involved in pregnancy and can stimulate or inhibit cell growth, induce chemotaxis of cells, and regulate cell differentiation and the expression of other cytokines. In the cytokine–cytokine receptor interaction in cluster 2, *INHA*, *IL33*, *IL12*, *RB1*, *GDF5*, *LTA*, *CXCR6*, *IL2RA*, *CSF2RB*, *IL17C*, *TNFRSF9*, *IL7R*, *BMP15*, *CCL22*, *IL25*, *CCR4*, *CCL17*, *TNFSF15*, *CX3CR1*, *TNFSF14*, *TNFSF8*, *TNFSF13B*, *TNFSF13*, *IL20RA*, *CSF3R*, *IL11*, *CXCL14*, and *TNFRSF1B* were chosen and visualized using a heatmap (Figure 6b). More importantly, *CCL17*, *CCL22*, and *CCR4* are structurally related molecules that regulate cell trafficking of various types of leukocytes and play a fundamental role in T-cell development, homeostasis, and the immune system [41]. Both *TNFSF9* and the *IL* family also contribute to regulating the function of immune cells and play an anti-tumor role [42]. *CXCL14* and *CSF3R* play major roles in the growth and differentiation of granulocytes [43]. This led us to assume that the downregulation of these genes is closely related to the activation of leukocytes required for ovulation and immune response related to pregnancy preparation during the late metestrus and diestrus periods (D06–D09).

#### 3.2.3. Cell Proliferation and Growth in Luteal Phase

The treemap showed negative regulation of interleukin-6 production as the highly enriched processes in cluster 3 (Figure 7a). Other biological processes enriched in cluster 3 included the morphogenesis-related terms such as negative regulation of the BMP signaling pathway, skeletal system morphogenesis, and embryonic forelimb morphogenesis (Figure 7b). In the KEGG bar plot of cluster 3, embryo development-related pathways (the hedgehog signaling pathway and osteoclast differentiation) and immune system-related terms (complement and coagulation cascades, *Staphylococcus aureus* infection, and basal cell carcinoma) were observed as the major pathways (Figure 7c). Considering the BP term (negative regulation of interleukin-6 production, negative regulation of BMP signaling pathway, skeletal system morphogenesis, and embryonic forelimb morphogenesis), we focused on the hedgehog signaling pathway among the KEGG pathways (Figure 8a). In this pathway, the displayed DEG were downregulated on D06 (Figure 8b).

The hedgehog signaling pathway affects embryonic development and morphogenesis and is known to be involved in follicular growth and development in mouse and *Drosophila* ovaries [44]. The current study revealed that *PTCH2*, *GLI1*, *GLI2*, *ARRB1*, *ARRB2*, *HHIP*, *LRP2*, *GRK3*, and *IHH* were significantly downregulated in the hedgehog signaling pathway. Among them, *IHH*, *GLI1*, and *HHIP* were identified as the noticeable DEG in the late metestrus stage because of their high expression levels. *IHH* is involved in cell proliferation and differentiation, especially bone cell development [45]. The GLI protein, encoded by the *GLI* gene, acts as a factor for hedgehog signaling, determining the fate of many types of cells and most organs during embryonic development and contributing to cell proliferation and pattern formation [46]. Similarly, *HHIP* is known as a gene involved in cell proliferation, growth, and morphogenesis [47]. Among the DEG of the hedgehog signaling pathway, studies have shown that transcription factors *GLI1* and *HHIP* are expressed in the CL and act as either transcriptional activators or repressors [48]. This pathway was downregulated just after the follicles ovulated and before the CL began to develop in earnest. This was because the adequate differentiation of ovarian cells and normal follicular development are essential for ovulation and subsequent CL formation [49,50]. Thus, this pathway reflects when both follicular growth and CL development have occurred. Therefore, it can be postulated that the DEG in the hedgehog signaling pathway involved in cell growth and proliferation is downregulated because it affects the development of follicular growth in the ovary.

## 4. Conclusions

In conclusion, this study aimed to show the dynamic changes in gene expression that take place in the ovary of the nonpregnant gilt during the estrous cycle. According to the results of the transcriptome analysis, the gene expression wave was more active during the luteal phase than in the follicular phase. Our results showed that the late metestrus and diestrus periods exhibited a remarkable expression pattern, and we confirmed that these periods are important to prepare for pregnancy or for the next estrous cycle after ovulation. Therefore, it is necessary to understand the mechanisms of ovarian regulation during this period and study the relationship between pregnancy and estrous cycles that may occur later. Our study suggests a promising basis for the understanding of successful reproduction at the genomic level in the porcine industry and is an informative resource for further studies on the porcine ovary.

## Figures and Tables

**Figure 1 animals-12-00376-f001:**
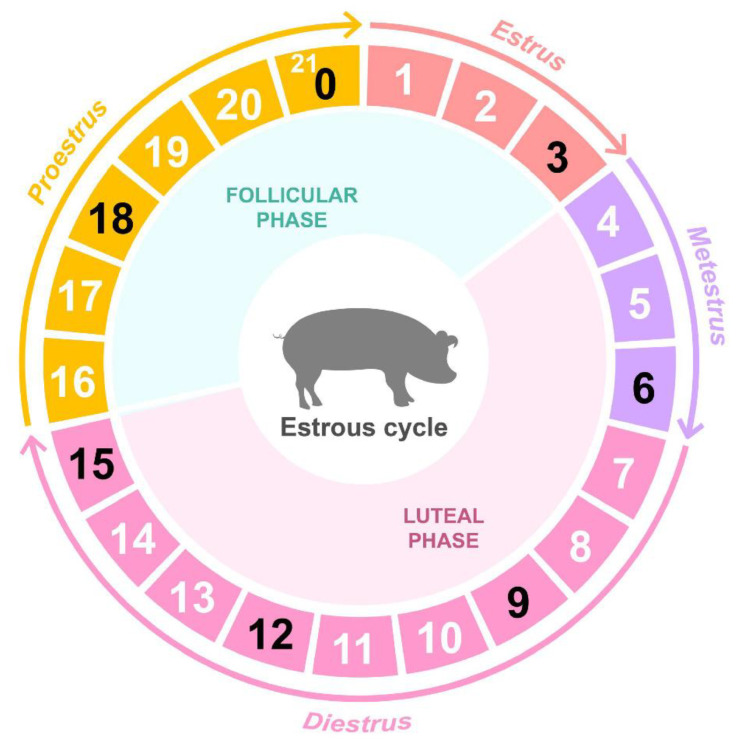
Overview of ovary sampling throughout the porcine estrous cycle. Stages of the estrous cycle in porcine ovary at seven time points (D00, D03, D06, D09, D12, D15, and D18).

**Figure 2 animals-12-00376-f002:**
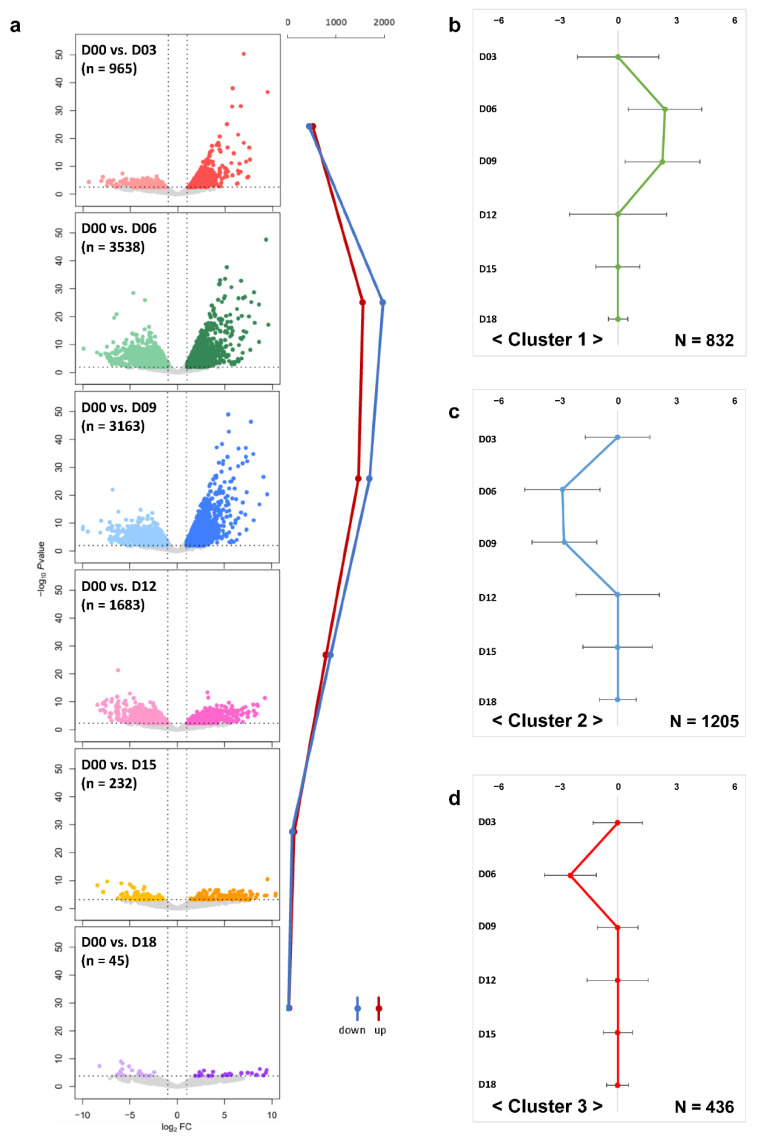
Dynamic changes of differentially expressed genes (DEG) during the porcine estrous cycle. Dynamic views and expression patterns of DEG using a volcano plot and k-medians clustering algorithm in Multi Experiment Viewer (MeV). (**a**) Comparing the start of estrous cycle (D00) with each time point (D03, D06, D09, D12, D15, and D18). The x and y axes of the volcano plots were scaled using log_2_ fold changes and −log_10_ *p*-value. Significant DEG (false discovery rate < 0.05; log_2_FC ≤ |1|: total, log_2_FC ≥ 1: up, log_2_FC ≤ −1: down; the total number of DEG: n) in the volcano plots are represented by six different colors corresponding to each time point. The line graph reveals the changes in the number of DEG (up, down) for each time point. (**b**–**d**) Gene clustering analysis disclosed three clear expression patterns for each time point (N: the number of DEG against each cluster).

**Figure 3 animals-12-00376-f003:**
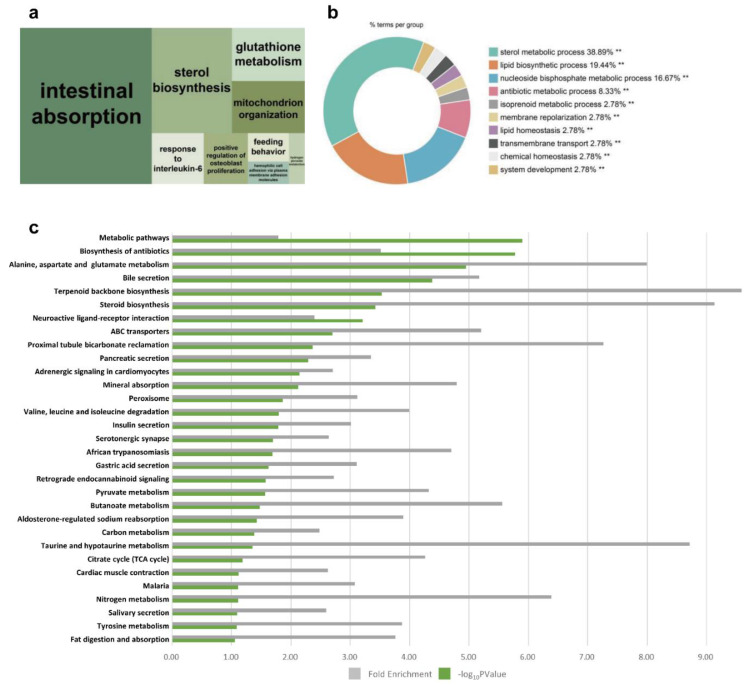
Visualization of functional analyses of cluster 1. (**a**) A biological process (BP) treemap was constructed using REVIGO. (**b**) BP pie graph using Cytoscape (** *p* < 0.05). (**c**) Bar graph of Kyoto Encyclopedia of Genes and Genomes (KEGG) pathways.

**Figure 4 animals-12-00376-f004:**
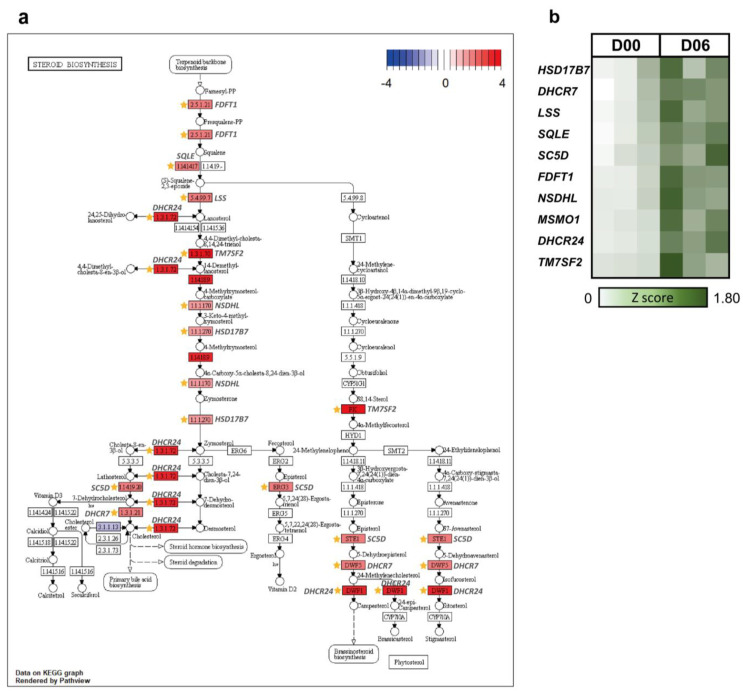
Pathway-based expression information including DEG of cluster 1. (**a**) Log_2_FC change in the genes of the steroid biosynthesis (KEGG) using clusterProfiler at D06 and DEG in the pathway were marked. (**b**) Heatmap of selected DEG on comparing D06 with D00.

**Figure 5 animals-12-00376-f005:**
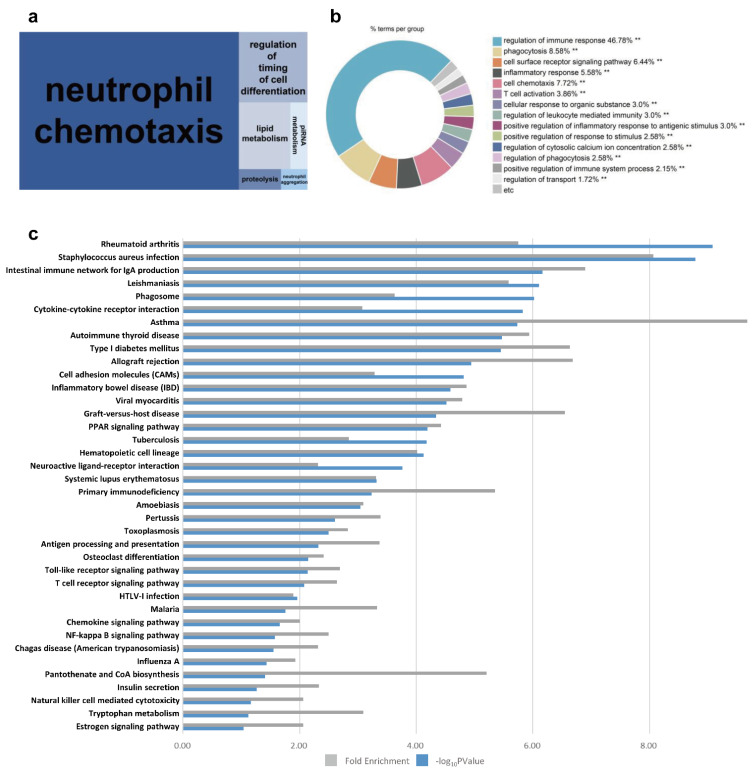
Visualization of functional analyses of cluster 2. (**a**) A BP treemap was constructed using REVIGO. (**b**) BP pie graph using Cytoscape (** *p* < 0.05). (**c**) Bar graph of KEGG pathways.

**Figure 6 animals-12-00376-f006:**
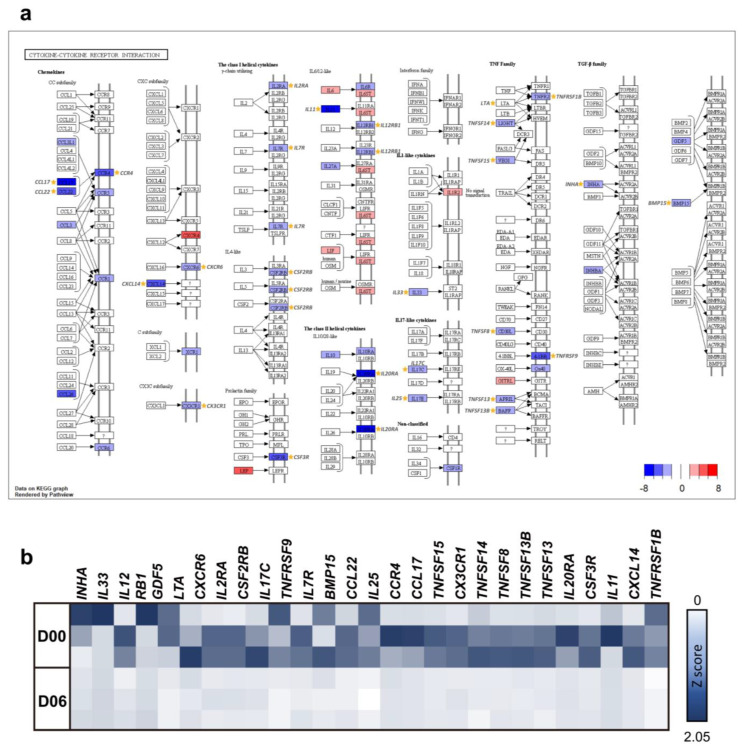
Pathway based expression information including DEG of cluster 2. (**a**) Log_2_FC change in the genes of the cytokine–cytokine receptor interaction (KEGG) using clusterProfiler at D06 and DEG in the pathway were marked. (**b**) Heatmap of selected DEG on comparing D06 with D00.

**Figure 7 animals-12-00376-f007:**
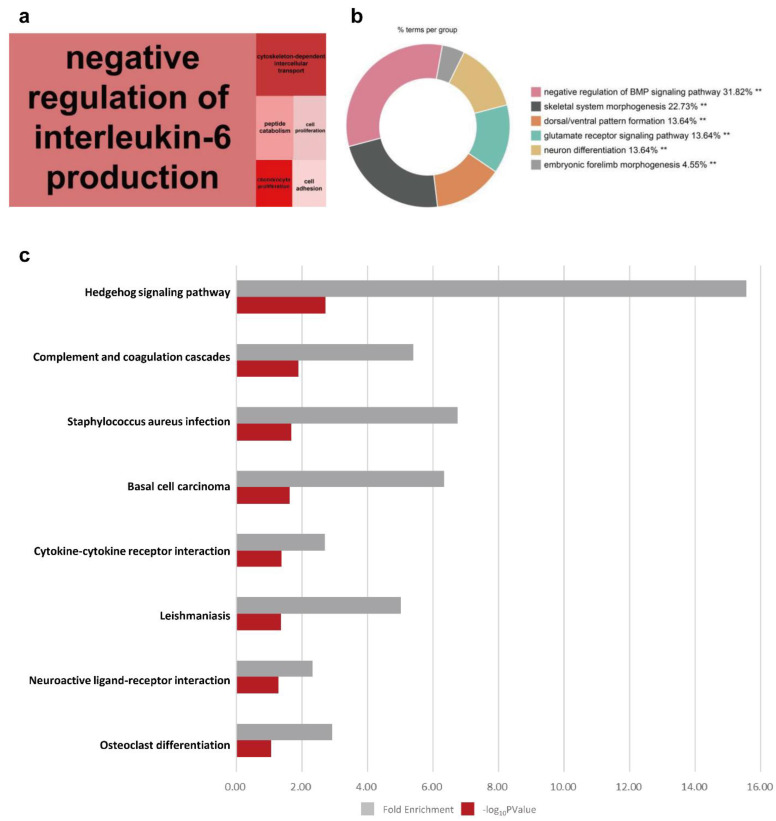
Visualization of functional analyses of cluster 3. (**a**) A BP treemap was constructed using REVIGO. (**b**) BP pie graph using Cytoscape (** *p* < 0.05). (**c**) Bar graph of KEGG pathways.

**Figure 8 animals-12-00376-f008:**
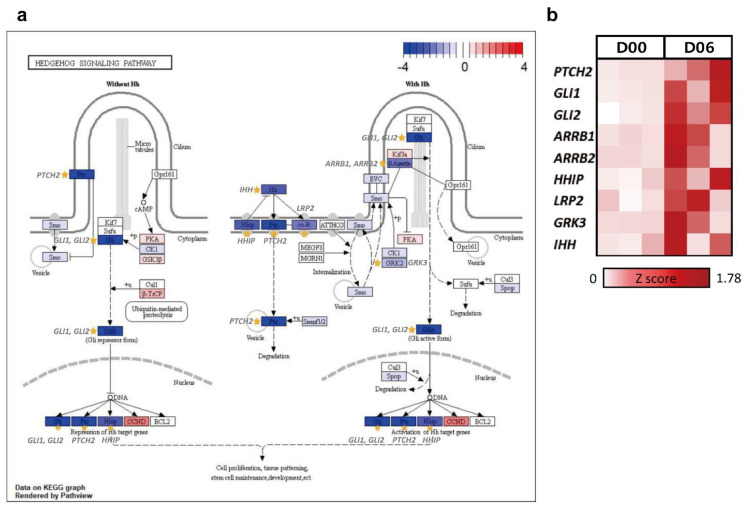
Pathway-based expression information including DEG of cluster 3. (**a**) Log_2_FC change in the genes of the hedgehog signaling pathway (KEGG) using clusterProfiler at D06 and DEG in the pathway were marked. (**b**) Heatmap of selected DEG on comparing D06 with D00.

**Table 1 animals-12-00376-t001:** Summary of the RNA-sequencing results and mapped reads alignment of porcine ovarian samples obtained during the estrous cycle.

Group	Sample	Raw Reads	Clean Reads Rate (%)	Uniquely Mapped Reads Rate (%)	Overall Alignment Rate (%)
Day 0	D00C-Ovary-1	19,596,597	16,133,880 (82.33)	95.12	98.36
	D00C-Ovary-2	19,851,794	16,852,302 (84.89)	94.80	98.31
	D00C-Ovary-3	19,119,940	16,298,225 (85.24)	94.84	98.49
Day 3	D03C-Ovary-1	18,523,866	15,737,437 (84.96)	94.72	98.39
	D03C-Ovary-3	19,201,088	16,311,360 (84.95)	94.83	98.46
Day 6	D06C-Ovary-1	19,286,199	16,446,174 (85.27)	94.93	98.56
	D06C-Ovary-2	20,334,817	17,479,844 (85.96)	94.46	98.50
	D06C-Ovary-3	21,255,422	18,276,501 (85.99)	94.07	98.30
Day 9	D09C-Ovary-1	19,901,123	17,088,002 (85.86)	94.31	98.42
	D09C-Ovary-2	21,040,934	18,103,889 (86.04)	94.10	98.45
	D09C-Ovary-3	18,146,293	15,539,159 (85.63)	94.47	98.45
Day 12	D12C-Ovary-1	19,069,936	16,168,172 (84.78)	94.84	98.45
	D12C-Ovary-2	18,537,734	15,955,652 (86.07)	93.39	98.28
	D12C-Ovary-3	18,664,749	16,005,643 (85.75)	93.50	98.00
	D12C-Ovary-4	17,946,020	15,377,254 (85.69)	94.05	98.32
Day 15	D15C-Ovary-1	18,858,268	15,953,501 (84.60)	94.57	98.32
	D15C-Ovary-2	18,255,019	15,646,924 (85.71)	94.56	98.34
	D15C-Ovary-3	19,733,962	16,849,829 (85.38)	94.55	98.28
	D15C-Ovary-4	20,762,176	17,717,261 (85.33)	94.47	98.17
Day 18	D18C-Ovary-1	22,291,469	19,003,553 (85.25)	94.67	98.31
	D18C-Ovary-2	19,663,920	16,727,802 (85.07)	94.88	98.33
	D18C-Ovary-3	19,968,688	17,000,527 (85.14)	95.20	98.36

## Data Availability

The data generated in this study are deposited in the NCBI SRA data base under accession no. SRP127622.

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
