# Peer review of "Time Series Ovarian Transcriptome Analyses of the Porcine Estrous Cycle Reveals Gene Expression Changes during Steroid Metabolism and Corpus Luteum Development"

_animals, 2022, doi:10.3390/ani12030376_

Round 1
Reviewer 1 Report
This study on time series ovarian transcriptome analyses of the porcine estrous cycle during corpus luteum development is an important area considering the critical role of ovarian physiology in reproductive efficiency of pig farming; however, the manuscript needs major revision especially English language editing and contents of results and discussion part.
Abstract: It needs to be re-written. It should mention the background, objectives, methodology, results and conclusion of the study.
Line 45: Authors should either use the term “ovary” of “the female gonad”
Line 46: Use the term “female reproductive organ” instead of “gland”
Line 63-65: Provide the citation for these studies.
Materials and methods: Functional validation of DEG is missing. Authors should provide an information on this aspect of experiment.
Remove “representative reproductive organ” and Re-write the line 147-148 as: Ovary is the female reproductive organ which undergoes functional and physiological changes during each estrous cycle.
Table 1: There should be uniformity in nomenclature. Like Day 0- Day 9 group mentions: D0C-Ovary- D9C-Ovary but rest of the groups follow the pattern D12C-Ovary-1 to D12C-Ovary-4
Figure 2a. The volcano plot shot be marked with the DEG analysis group like D00 vs. D03 or D00 vs. D09
Line 168 -169: Discuss this point with respect to physiological stage of the study like D06 and D09.
Line 170-172: Modify this sentence: Furthermore, a large proportion of down regulated DEG found on D06corresponding to the metestrus, D09 and D12 corresponding to the diestrus phase were identified (D06: 55.90 %; D09: 53.68 %; D12: 52.88 %).
Line 174-175: Rewrite as “……the DEG were clustered into three clusters………”
Line 179-182: Discuss these in detail citing references from suitable studies in pigs and other species. Authors can refer to the transcriptional changes during luteal stage in porcine reported in the recent study. “Transcriptome profiling of different developmental stages of corpus luteum during the estrous cycle in pigs. Genomics, 113(1), pp.366-379”.
Line 201-202 Heatmaps were drawn to visualize specific DEG, based on D06, when the most dynamic change occurred in all three clusters (Figs. 3–5). Discuss the possible physiological reasons for this citing recent study in different species including pigs.
Line 206-207: Rewrite it as “As indicated in cluster 1…”
Line 211-212: These significant terms were consistent with the changes that occur in hormones involved in the estrous cycle. Discuss the possible physiological reasons for this citing recent study in different species including pigs.
Line 2019-222: “Previous studies have shown that rat androstenedione synthesis maintains the function of the CL and increases progesterone biosynthesis [25]. It has also been shown that androgens can maintain luteal function in mice lacking the classic progesterone receptor [26]. Therefore, it can be concluded that various steroid hormones are involved in steroid biosynthesis and progesterone formation in the CL.” Rewrite these points with relevant studies and references supporting the findings of the present study.
Line 222-223: Rewrite as “In cluster 1, the fold change in the expression level of 10 steroid biosynthesis DEG…”
Line 225-226: “HSD17B7 completes the molecular cloning of all genes of the cholesterol biosynthesis pathway [27].” Rephrase this sentence to make it more meaningful and relevant.
Line 232: Replace the word deeply with closely
Line 233-234: Decreased blood flow in the uterus and the effect of hormones decreases the quality of oocytes, which causes infertility. Effect of which hormones and how? Rephrase this sentence to make it more meaningful with physiologically relevant explanation.
Line 235-237 and Line 346 in Conclusion: Diestrus (D06–D09), corresponding to all clusters, are the periods when the CL development is most active, resulting in increased concentrations of progesterone. Here D06 is mentioned in Diestrus phase whereas, in Fig. 1 D06 is covered in metestrus. Kindly co-relate and correct.
Line 246-247: Thus, progesterone synthesis by CL affects luteinization and luteolysis, as well as CL maintenance and structure due to pregnancy [26].Rephrase this sentence to make it more meaningful and relevant.
Line 257: Remove “Similarly, in the pie graph” and begin with “Immune-related terms, such………..”
Line 264: Replace “are seen in the bar plot of cluster 2” with “were enriched in cluster 2”
Line 266-267: There are several KEGG orthologues related to cytokine–cytokine receptor interaction; one-third of these are expressed. In Figure 4d, most of the downregulated genes were DEG. Rephrase this sentence to make it more meaningful and relevant.
Line 269: Replace “with the sperm from outside” with “with the sperm within 2–3 d resulting into establishment of pregnancy”
Line 270-274: Rephrase this sentence to make it more meaningful and relevant.
Line 273-274: This can be postulated that the organs of the same reproductive system are organically connected, so the ovary is also affected by the oviduct through which sperms enter.
Line 303-304: Replace “Finally, the treemap for cluster 3 showed negative regulation of interleukin-6 production as the largest part” with “The treemap for cluster 3 showed negative regulation of interleukin-6 production as the highly enriched process in cluster 3”
Line 304-306: Rephrase as “Other biological process enriched in cluster 3 included the morphogenesis-related terms such as negative regulation of BMP signaling pathway, skeletal system morphogenesis and embryonic forelimb morphogenesis.
Line 310-315: Considering the BP term (negative regulation of interleukin-6 production, negative regulation of BMP signaling pathway, skeletal system morphogenesis, and embryonic forelimb morphogenesis), we focused on the hedgehog signalling pathway among the KEGG pathways (Figure 5d). Is there any correlation between hedgehog signalling pathway and the BP terms considered. Suitable references should be cited or reason should be explained. Then it is mentioned “In this pathway, the remarked DEG are down regulated on D06 and can be checked in the heat map compared with that in the control (Figure 5e).Rephrase this sentence to make it more meaningful with physiologically relevant explanation. “can be checked in the heat map” these kind of usage should be avoided.
Line 326-327: “Among them, IHH, GLI1, and HHIP were identified as the noticeable DEG because of their high expression levels during*”. *Mention the stage of estrous cycle.
Line 334-339: “This pathway was down regulated just before the follicles ovulated and before the CL began to develop in earnest.” This sentence is vaguely written because only after ovulation CL development occurs. Correct accordingly. The next line mentions “Thus, this pathway reflects when both follicular growth and CL development have occurred.” It needs to be correlated for making it meaningful. “Therefore, it can be postulated that the DEG in the hedgehog signalling pathway involved in cell growth and proliferation is down regulated because it affects the development of follicular growth in the ovary.” What does the authors want to convey? Proper logic is missing. References from suitable studies need to be added.
Was any hormonal assay conducted to demarcate the different stages of estrous cycle and complement the findings of the study? If not. It should be mentioned as one of the limitations of the study while discussing the results.
Conclusion: Rewrite it. Should be short and crisp mentioning salient findings of the study.
Line 344: Remove this line “The changes occurring in the ovary were investigated through functional analyses.”
Line 345-348: Remove these lines. Already mentioned in results and discussion.
Line 352-345: Remove these lines. References in conclusion should be avoided.
Reviewer 2 Report
This paper entitled "Time Series Ovarian Transcriptome Analyses of the Porcine Estrous Cycle Reveals Gene Expression Changes during Steroid Metabolism and Corpus Luteum Development" presents a study describing the changes at the transcriptomic level in ovarian tissue throughout the estrous cycle in the sow.
Apparently, one of the problems with the paper is that it is based on previously published data (DOI:10.1038/s41598-018-23655-1), however, as the authors point out in the introduction, the approach used to discuss the results focuses only on ovarian tissue, unlike the other paper.
Although the article presents interesting information, there are some aspects that should be clarified beforehand:
L-77: In the material and methods, reference is made to the use of a total of 22 sows, however it is not indicated how many were used per day. In one of the supplementary figures (Fig. S1), it appears that on some days 3 sows were used while on other days 4 were used.
L-80: I think it would be more appropriate to refer to it as the first day of estrus behavior.
L-89: For the RNA extraction, was any methodology followed to take the samples from the different ovarian structures? I am referring to whether the stroma, corpora lutea in the luteal phase or follicles during proestrus were discriminated.
L-91: I think it would be interesting to include a range of the RIN value obtained.
L-122: Review the abbreviations for GO and KEGG, as they are then reintroduced in the discussion section.
L-124: It is necessary to include the term biological process before BP, however the abbreviation bp is already used to refer to base pairs. It would be necessary to modify some of them, or I think it would be simpler to eliminate the second one.
L-143: Was any validation of the results obtained by q-PCR performed?
L-182: From this point on, the results focus mainly on the luteal phase, with subsequent analyses being performed only on the comparison between day 0 and day 6. Although the reason for this is highlighted in the conclusions, I believe that at this point it should be introduced in some way that the subsequent results focus on this comparison.
L-184: In order to visualize the common and unique DEGs for each of the comparisons, it would be interesting to present a Venn diagram among the 6 comparisons. This would contribute to the visualization of the results presented in section 3.1.
L-186-194: I believe this entire section corresponds to the caption of figure 2.
L-203: In sections 3.2.1 and 3.2.2 the term luteal phase is used to further analyze the changes between day 0 and day 6, but in section 3.2.3 it is changed to early luteal phase to refer to the same day. I believe that the criteria should be unified to avoid confusion, and taking into account that comparisons are also made on other days within the luteal phase, it would be more accurate to speak of the early luteal phase.
L-212: Why are the terms related to the digestive system consistent with the changes that occur in hormones involved in the estrous cycle?
L-232-236: How is this blood flow information related to the results obtained? Were any angiogenic or similar factors found to be affected?
L-238-248: I believe that all this part is not directly related to the results obtained. It is an explanation of the physiology of progesterone at the reproductive level. While it is true that the results support an increase in steroid production on day 6, I think the explanation should focus on whether these genes follow a shift from estrogen production to progesterone production.
L-267: “In Figure 4d, most of the downregulated genes were DEG”. The abbreviation DEG includes the term genes, so this sentence should be corrected, noting that most DEGs were downregulated. The authors should revise this as it also occurs in some other section.
L-325: “The current study revealed that PTCH2, GLI1, GLI2, ARRB1, ARRB2, HHIP, LRP2, GRK3, and IHH were significantly involved in the hedgehog signaling pathway”. I consider that this sentence is not correct, since what the work reflects is that these transcripts were downregulated on day 6 with respect to day 0.
L-334: “This pathway was downregulated just before the follicles ovulated and before the CL began to develop in earnest.” What comparisons do you mean by just before ovulation and just before CL formation? Day 0 vs 3 and day 0 vs 6 respectively? This is a bit confusing, already 3 days after the onset of estrus they have probably ovulated.
-The heat maps in Figures 3, 4 and 5 should include a legend reflecting the intensity of the expression in the number values used.
-The supplementary figures do not have captions. In addition, the first one, which I believe is a principal component analysis, would be interesting to include in the main text.
Reviewer 3 Report
The manuscript evaluated the transcriptional patterns of porcine ovaries at 7 time points. The authors produce a gene catalogue which might be of interest in improving reproduction traits in pigs. However, I find the hypothesis for the study lacking in the Abstract and Introduction sections. Main drawback is that phenotypes are missing, e.g. hormones or histological evaluation have not been analysed. Especially blood parameters would be important to support the ethological and molecular observations and to show standardized sampling. The description of the sampling material is rather poor. Authors need to show more effort in visualize & interpret their data.
Simple summary:
- I suggest to skip the first sentence.
- The luteal phase and the follicular phase should be shortly introduced with regards to their biological impact.
Abstract:
- Line 20: “results in changes”: Please specify. That is too vague.
- Line 23: “identification”: in fact, experiments like this should provide new hypotheses.
- Lines 27-33: Here the connection to biology is not present. There is no need for the claim that clusters 2 and 3 are downregulated.
Introduction:
- Line 66: “understanding”: I think the results refer just to a catalogue of DEGs.
- Special anatomical features in the pig are missing.
M&M:
- Lines 77-84: What is the relationship of the studied animals? How was "at least two normal duration estrous cycles" observed? What about the methods and parameter used to collect (standardize) the ethological data? How did the slaughter process and sampling take place (time interval, sampling duration etc.)? What anatomical landmarks were used for reproducibility?
- Lines 86: “sampling” instead of “transcriptomes”. The caption should include the rationale for the sampling times.
- Lines 117 & 121: cut-offs for the FCs are inconsistent. Why?
- Authors used DAVID as annotation tool. It has its merits, but the last update is already from 2016.
- Lines 141-142: Please explain.
- The statistics do not include information on the consideration of relationship (e.g. father).
- What about alternative software tools for time analysis – R packages are available.
Results & Discussion:
- Line 161: Table 1 could be shifted to the Appendix section. Why sampling time points are not balanced? This weakens the comparison D0 vs D3.
- Figure 2A could be shifted to the Appendix section as well.
Round 2
Reviewer 2 Report
Reviewer 2
This paper entitled "Time Series Ovarian Transcriptome Analyses of the Porcine Estrous Cycle Reveals Gene Expression Changes during Steroid Metabolism and Corpus Luteum Development" presents a study describing the changes at the transcriptomic level in ovarian tissue throughout the estrous cycle in the sow.
Apparently, one of the problems with the paper is that it is based on previously published data (DOI:10.1038/s41598-018-23655-1), however, as the authors point out in the introduction, the approach used to discuss the results focuses only on ovarian tissue, unlike the other paper.
Although the article presents interesting information, there are some aspects that should be clarified beforehand:
Our response: We completely agreed with your critical concern. As you cited the already published data (Kim, Park et al. 2018), we previously studied using these materials to integrate different kinds of female reproductive tissues and identify the core regulation module in the integrative gene expression networks. However, we could not focus on independent ovarian transcriptomic changes and define the dynamic regulatory gene modules in ovary following the estrous cycle. Ovarian morphological phenotype changes have been already well described as we mentioned in the manuscript. However, the gene expression regulation through the estrous cycle has not been clearly characterized yet. We believe one reason for the lack is the difficulty of sampling reproductive tissues after sacrifice on each time point to consider the entire estrous cycle, and highly advanced integrative transcriptome analysis is essentially required to precisely characterize transcriptomic changes at once. In the current study, we have carefully focused on ovarian transcriptome integration analysis during the estrous cycle and identified gene expression regulation mechanisms regarding the cycle rolling. Please understand the use of ovarian data set and the current companion further study aims.
Reviewer's response:
I understand the use of the data. However, I believe that such facts should be stated, as you do, by properly citing such previous studies and contextualizing them appropriately in the final paper, in order to avoid confusion about the originality of the data.
L-77: In the material and methods, reference is made to the use of a total of 22 sows, however it is not indicated how many were used per day. In one of the supplementary figures (Fig. S1), it appears that on some days 3 sows were used while on other days 4 were used.
Our response: We sincerely appreciate your reviews and suggestions. This information can be checked through Table 1 and there are 4 samples on days 12 and 15. For more effective information delivery, we have added the number of samples in supplementary figure 1 [Page 5, Table 1, and Supplementary figure 1].
Reviewer's response:
My mistake, effectively you can check the information in that table.
L-80: I think it would be more appropriate to refer to it as the first day of estrus behavior.
Our response: Thanks to the reviewer’s suggestion, we have switched the word to “the first day” [Page 2, Line 85].
Reviewer's response:
I think there has been an error of interpretation, the first revision I was referring to in line 80-81, in the sentence "the first day of the behavior was designated as day 0.", be replaced by the first day of the estrus behavior... or the first day of behavioral change...
L-89: For the RNA extraction, was any methodology followed to take the samples from the different ovarian structures? I am referring to whether the stroma, corpora lutea in the luteal phase or follicles during proestrus were discriminated.
Our response: We truly understood your concern about RNA extraction and the procedure of sampling. In our study, the ovary was exposed by midventral laparotomy, and we collected a whole ovary on one side. During the process, the stroma, corpora lutea in the luteal phase of follicles during proestrus were not precisely distinguished.
Reviewer's response:
Therefore, the extracted RNA came from a pool of the different structures?
L-91: I think it would be interesting to include a range of the RIN value obtained.
Our response: We really appreciate your useful advice. Our data for which total RNA integrity was confirmed using an Agilent Technologies 2100 Bioanalyzer with a RIN value of 7 or higher was used. We thought that our data is reliable to proceed with the subsequent analyses and we added the contents of this information thanks to your comment [Page 3, Line 95-96].
Reviewer's response:
I also consider this to be an acceptable value.
L-122: Review the abbreviations for GO and KEGG, as they are then reintroduced in the discussion section.
Our response: As per the reviewer’s thankful advice, we have added the full name of GO and KEGG in the revised manuscript [Page 4, Line 127-128].
Reviewer's response:
Fixed.
L-124: It is necessary to include the term biological process before BP, however the abbreviation bp is already used to refer to base pairs. It would be necessary to modify some of them, or I think it would be simpler to eliminate the second one.
Our response: As you suggested, we have written down “biological process” before BP and also written down “base pair” for bp [Page 4, Line 130].
Reviewer's response:
Ok.
L-143: Was any validation of the results obtained by q-PCR performed?
Our response: We understood your concern. In recent days, RNA-seq research results are widely used without any experimental validation, otherwise, the qPCR for only a few selected genes are not necessarily required in general. Therefore, this study focused on functional validation analysis using multiple bioinformatics tools.
Reviewer's response:
I do not fully agree with this statement. It is true that many articles using RNA-seq are not validated by another complementary technique, however, I am of the opinion that this is justified when working with very valuable samples with a low amount of RNA in the extraction. I believe that this is not the case for these samples since working with tissue, the amount of RNA extracted is more than enough to perform both techniques. In any case, I understand that in the current work this type of validation has not been performed with a complementary technique.
L-182: From this point on, the results focus mainly on the luteal phase, with subsequent analyses being performed only on the comparison between day 0 and day 6. Although the reason for this is highlighted in the conclusions, I believe that at this point it should be introduced in some way that the subsequent results focus on this comparison.
Our response: Your concern is fully understood. In our study, the result of clustering analysis was found to be significant at D06 in all clusters. So, we focused on the luteal phase, especially on D06, and proceeded with the subsequent analysis [Page 6, Line 193-195].
Reviewer's response:
I am unable to find any sentence that clarifies that the subsequent analyses were done only with the comparison on day 6. Those lines belong to the caption of figure 2.
L-184: In order to visualize the common and unique DEGs for each of the comparisons, it would be interesting to present a Venn diagram among the 6 comparisons. This would contribute to the visualization of the results presented in section 3.1.
Our response: As your suggestion, we have included the upset plot instead of Venn diagram to visualize the common and unique DEGs among the 6 comparisons and added the contents in section 3.1. [Page 6, Line 181-183, and Supplementary figure 2].
Reviewer's response:
There must be some error with the lines you indicate, as they do not match the corrected version. In this case I suppose you are referring to L-177. However, you indicate that they share 11167 DEGs, is this correct? Supplementary files cannot be opened.
L-186-194: I believe this entire section corresponds to the caption of figure 2.
Our response: As your suggestion, these sentences are the legend of figure 2 and we have fixed the location thanks to the reviewer’s comment [Page 7, Figure 2].
Reviewer's response:
Ok.
L-203: In sections 3.2.1 and 3.2.2 the term luteal phase is used to further analyze the changes between day 0 and day 6, but in section 3.2.3 it is changed to early luteal phase to refer to the same day. I believe that the criteria should be unified to avoid confusion and taking into account that comparisons are also made on other days within the luteal phase, it would be more accurate to speak of the early luteal phase.
Our response: Thanks to the reviewer’s kind advice, we have changed the title of 3.2.3 which was “Early Luteal Phase” with “Luteal Phase” to avoid the confusion [Page 12, Line 339].
Reviewer's response:
Ok, I think it would have been more correct to call them all as early luteal phase (D6) considering that the rest of the days correspond to more advanced moments of the luteal phase. However, it is now unified.
L-212: Why are the terms related to the digestive system consistent with the changes that occur in hormones involved in the estrous cycle?
Our response: Thank you for your helpful question. Digestive system-related terms are expressed in a similar way with hormones involved in the estrous cycle because this result is from the same clustered DEGs which is corresponding with hormone secretion. Also, we changed the terms from “digestive system related” to “secretory and digestive system” for better understanding [Page 8, Line 230-231].
Reviewer's response:
Ok.
L-232-236: How is this blood flow information related to the results obtained? Were any angiogenic or similar factors found to be affected?
Our response: To support the sentence “Decreased blood flow in the uterus and the effect of hormones decreases the quality of oocytes, which causes infertility.”, we have included an additional reference (Nargund, Bourne et al. 1996) that shows oxygen tension and ATP concentration affect oocyte quality besides angiogenic [Page 9, Line 261-263].
Reviewer's response:
Ok.
L-238-248: I believe that all this part is not directly related to the results obtained. It is an explanation of the physiology of progesterone at the reproductive level. While it is true that the results support an increase in steroid production on day 6, I think the explanation should focus on whether these genes follow a shift from estrogen production to progesterone production.
Our response: We really appreciate your advice and as you mentioned, this part is supporting the results of D06 which increase in steroid production. In addition, we included your suggestion as “From this, it can be postulated that the expression of genes selected as DEG during this period has consequences related to the development of CL at the time and the resulting increase in progesterone concentration.” [Page 9, Line 266-269].
Reviewer's response:
Ok.
L-267: “In Figure 4d, most of the downregulated genes were DEG”. The abbreviation DEG includes the term genes, so this sentence should be corrected, noting that most DEGs were downregulated. The authors should revise this as it also occurs in some other section.
Our response: Thanks to the reviewer’s suggestion, we have switched the part to “most of the downregulated genes were identified as DEG.” [Page 11, Line 301-303].
Reviewer's response:
I was referring to the fact that the term genes is repeated twice, since it is included in the abbreviation DEG.
L-325: “The current study revealed that PTCH2, GLI1, GLI2, ARRB1, ARRB2, HHIP, LRP2, GRK3, and IHH were significantly involved in the hedgehog signaling pathway”. I consider that this sentence is not correct, since what the work reflects is that these transcripts were downregulated on day 6 with respect to day 0.
Our response: Your concerns are fully understandable. Even if the corresponding genes are downregulated, they could be involved in the hedgehog signaling pathway. This is because the corresponding genes are included in the pathway, which does not mean that those genes are not related to the pathway, but the function that the pathway performs is expressed differently [Page 14, Line 362-364].
Reviewer's response:
I think I expressed myself badly. I meant that this paper does not prove that the genes belong to such a pathway, therefore that sentence should be corrected.
L-334: “This pathway was downregulated just before the follicles ovulated and before the CL began to develop in earnest.” What comparisons do you mean by just before ovulation and just before CL formation? Day 0 vs 3 and day 0 vs 6 respectively? This is a bit confusing, already 3 days after the onset of estrus they have probably ovulated.
Our response: Thanks to your comments, we were able to find the mistake. It should be “This pathway was downregulated just after the follicles ovulated and before the CL began to develop in earnest.” and our intention was to mention that Day 6 is between the ovulation and CL formation [Page 14, Line 372-374].
Reviewer's response:
Fixed.
-The heat maps in Figures 3, 4 and 5 should include a legend reflecting the intensity of the expression in the number values used.
Our response: We have included the scale bar indicating the expression level beside the heatmap as your advice [Figure 3e, 4e, and 5e].
Reviewer's response:
The legend has been included but not the units in which it is expressed. Which units are 0 to MAX?
-The supplementary figures do not have captions. In addition, the first one, which I believe is a principal component analysis, would be interesting to include in the main text.
Our response: As suggested by the reviewer, we have added legends of supplementary figures in the main text [Page14, Line 399-407].
Reviewer's response:
Ok.
References
Kim, J.-M., J.-E. Park, I. Yoo, J. Han, N. Kim, W.-J. Lim, E.-S. Cho, B. Choi, S. Choi and T.-H. J. S. r. Kim (2018). "Integrated transcriptomes throughout swine oestrous cycle reveal dynamic changes in reproductive tissues interacting networks." 8(1): 1-14.
Nargund, G., T. Bourne, P. Doyle, J. Parsons, W. Cheng, S. Campbell and W. Collins (1996). "Associations between ultrasound indices of follicular blood flow, oocyte recovery and preimplantation embryo quality." Human Reproduction 11(1): 109-113.
Reviewer 3 Report
The origin of samples has to be described also in this manuscript, including e.g. the anatomical landmarks used.
M&M:
- Lines 77-85: The issues already raised need to be addressed. Please make the effort to summarize the information so that the reader of your paper gets enough details! (What is the relationship of the studied animals? How was "at least two normal duration estrous cycles" observed? What about the methods and parameter used to collect (standardize) the ethological data? How did the slaughter process and sampling take place (time interval, sampling duration etc.)? What anatomical landmarks were used for reproducibility?)
Author Response
The origin of samples has to be described also in this manuscript, including e.g. the anatomical landmarks used.
M&M:
- Lines 77-85: The issues already raised need to be addressed. Please make the effort to summarize the information so that the reader of your paper gets enough details! (What is the relationship of the studied animals? How was "at least two normal duration estrous cycles" observed? What about the methods and parameter used to collect (standardize) the ethological data? How did the slaughter process and sampling take place (time interval, sampling duration etc.)? What anatomical landmarks were used for reproducibility?)
Our response: Thank you for your comments. Based on your suggestions, we have included this information about the anatomical landmarks addressed in the previous response in the manuscript as “These sampling materials were used in the previous study to integrate different kinds of female reproductive tissues and identify the core regulation module in the integrative gene expression networks.” and cited the previous study once more [Page 2, Line 82-85 and 87-90].